


# 1 Tsunami damage to ports: Cataloguing damage to create fragility 2 functions from the 2011 Tohoku event

Constance Ting Chua [1,2], Adam D. Switzer [1,2], Anawat Suppasri [3], Linlin Li [4], Kwanchai Pakoksung [3],
David Lallemant [1,2], Susanna F. Jenkins [1,2], Ingrid Charvet [5], Terence Chua [1], Amanda Cheong [6] and Nigel
Winspear [7]
[1] Asian School of the Environment, Nanyang Technological University, Singapore
[2] Earth Observatory of Singapore, Nanyang Technological University, Singapore
[3] International Research Institute of Disaster Science, Tohoku University, Sendai, Japan
[4] School of Earth Sciences and Engineering, Sun Yat-Sen University, Guangzhou, China
[5] Formerly Department of Statistical Science, University College London, London, United Kingdom
[6] JBA Risk Management Pte Ltd, Singapore
[7] Formerly SCOR Global P&C, Singapore
*Correspondence to*: Constance Chua (CCHUA020@e.ntu.edu.sg)
**Abstract.** Modern tsunami events have highlighted the vulnerability of port structures to these high-impact but infrequent
occurrences. However, port planning rarely includes adaptation measures to address tsunami hazards. The 2011 Tohoku
tsunami presented us with an opportunity to characterise the vulnerability of port industries to tsunami impacts. Here, we
provide a spatial assessment and photographic interpretation of freely available data sources. Approximately 5,000 port
structures were assessed for damage and stored in a database. Using the newly developed damage database, tsunami damage
is quantified statistically for the first time, through the development of damage fragility functions for eight common port
industries. In contrast to tsunami damage fragility functions produced for buildings from existing damage database, our
fragility functions showed higher prediction accuracies (up to 75% accuracy). Pre-tsunami earthquake damage was also
assessed in this study, and was found to influence overall damage assessment. The damage database and fragility functions for
port industries can inform structural improvements and mitigation plans for ports against future events.



## 1. Introduction

Port assets are vulnerable to the physical damage caused by tsunamis and cascading effects such as extensive supply chain disruption. For example, transoceanic waves from the 2004 Indian Ocean tsunami resulted in heavy damage to maritime facilities across the Indian Ocean. On the west coast of Banda Aceh, Indonesia, all harbours and landing piers between Lhok Nga and Meulaboh were destroyed and unusable (Janssen, 2005) and across the Indian Ocean, heavy damage to maritime facilities reportedly resulted in the closure of Nagappattinam Port, India for weeks (Mahshwari et al., 2005). On the same note, the 2011 Tohoku (Great East Japan) tsunami caused damage to many ports along the Pacific coast in the Tohoku region. The affected ports suffered from a contraction in export and import values following the tsunami (March – May 2011) of 57.5% and 61.6% respectively, relative to the preceding 5-year average for the same period (Japan Maritime Centre, 2011). Total economic losses for tsunami damage to Japan's marine vessels, ports and maritime facilities were approximated at US$ 12 billion (Muhari et al., 2015). A recent study speculated that earthquakes greater than Mw 8.5 from the Manila-trench could result in the loss of functions in up to five major ports including Kaohsiung and Hong Kong (Otake et al., 2019). Additionally, threats from future tsunami events are expected to be exacerbated by rising sea levels (Li et al., 2018), which imply greater risks for port assets located near tsunami sources.

With about 80% of global trade volume carried by sea, ports are critical nodes in international trade. Ports are also home to industrial clusters and critical facilities such as manufacturing firms and power plants due to the convenience they provide. With increased seaborne trade, globalisation of complex industrial processes and dependence on ports for economic development, port areas are only expected to develop further. However, port planning rarely accounts for adaptation to natural hazards and coastal protection structures are usually built to mitigate short-term hazard scenarios such as coastal flooding and wave damage (Lam and Lassa, 2017).

Tsunamis are high-impact events but infrequent occurrences, which makes their potential impacts to ports difficult to quantify. The expected increase in the exposure of port assets to coastal hazards, combined with our limited experience with tsunamis in modern ports, demonstrates a clear need to better understand how port structures might respond to tsunami impacts.

Structural damage resulting from tsunami impacts has generated considerable interest since the 2004 Indian Ocean tsunami (e.g. Nistor et al., 2010, Leelawat et al., 2016; Song et al., 2017; Suppasri et al., 2019). Structural damage is most commonly quantified in the form of tsunami damage fragility functions. First developed for tsunami events by Koshimura et al. (2009), tsunami fragility functions express the probability that a structure exceeds a prescribed damage threshold for a given tsunami flow characteristic or intensity measure. Pioneering work in the development of tsunami fragility functions has been largely focused on damage to residential and commercial buildings (e.g. Leone et al., 2011, Reese et al., 2011; Mas et al., 2012; Gokon et al., 2014). In recent years, the study of tsunami structural fragility has been extended to critical infrastructure such as roads and bridges (Akiyama et al., 2014; Shoji and Nakamura, 2017; Williams et al., 2020).

Despite recent efforts, our understanding of tsunami impacts on ports still falls short. The coverage of tsunami-induced damage on port structures in existing literature is by and large limited to qualitative assessments. To date most studies on tsunami





structural damage to ports are in the form of post-tsunami surveys, which document damage observations and describe the
failure mechanisms of harbour elements such as breakwaters, quay walls and wharves (e.g. Meneses and Arduino, 2011; Fraser
et al., 2012; Hazarika et al., 2013; Paulik et al., 2019; Benzair et al., 2020) and port facilities such as oil tanks, cranes and
equipment (e.g. Scawthorn et al., 2016; Percher et al., 2013; Sugano et al., 2014). Some studies have attempted to reconstruct
structural impacts to port facilities by evaluating design specifications of structures or examining specific tsunami behaviour
such as bore impact linked to structural damage (e.g. Nayak et al., 2014; Kihara et al., 2015; Chen et al., 2016; Huang and
Chen, 2020). Though recent studies attempted to quantify tsunami damage to port facilities, the focus of these standalone
studies are specific to certain port industries, namely warehousing (Karafa et al., 2018) and fishery industries (Imai et al.,
2019), and therefore do not provide a comprehensive view of the damage sustained by different port industries. While
necessary for the improvement of structural design, efforts so far are not adequate in quantifying tsunami damage statistically.
This study serves as a starting point in characterising the vulnerability of port industries to tsunami impacts, through the
assessment and quantification of structural response to tsunami inundation depths. The objective of this study is two-fold – (i)
to develop a tsunami damage database for port structures impacted during the 2011 Tohoku tsunami, and based on the damage
database, (ii) to construct tsunami damage fragility functions for port industries. The 2011 Tohoku tsunami presents a unique
opportunity to study tsunami damage to port structures due to the extent and severity of damage, and the large ensemble of
data collected post-tsunami (Table 1). The combination of densely recorded tsunami flow measurements, well-documented
surveyed damage data and high-quality photographic evidence available offers an unparalleled resource for this research.
In this paper, we develop the first tsunami damage database for port industries and their related structures. We also present the
first sets of tsunami damage fragility models for common industries found in the port hinterland. We describe the data sources
and methods to develop this damage database, and demonstrate in detail how the damage database addresses limitations found
in past studies. Fragility functions are constructed by reviewing and employing best practices in the field. Unique to this work,
we also evaluated the robustness of tsunami fragility functions against the influence of pre-tsunami earthquake effects. We
conclude by highlighting some key application opportunities of this dataset and providing recommendations for overcoming
current limitations found in this study. This study provides a blueprint for translating post-event damage surveys into fragility
functions, which can be used to forecast future tsunami-induced damage to ports.
**2. Study site**
The northeast coast of Japan, also known as the Tohoku region, was severely impacted by the Tohoku tsunami on 11 March
2011 (Fig. 1). Port operations along the Pacific Coast in Tohoku and eastern Kanto regions were disrupted due to debris and
severe damage to buildings, loading facilities, wharfs, fuel facilities and seawalls (Takano et al., 2012). Damage patterns varied
along the Tohoku coastline. The Tohoku coastline is mainly coastal plains and ria coasts. Coastal plains are extensive areas of
low-lying flat terrain, while ria coasts, formed by submergence of former river valleys, typically have limited flat terrain. Ria
coasts are characterised by narrow funnel-shaped coastal inlets bounded by steep slopes such as mountains. In coastal plains,





damage severity transitioned gradually with distance inland, decreasing as inundation depths decrease with distance inland
(De Risi et al., 2017). In ria coasts, the spatial distribution of damage was uneven because flow characteristics i.e. velocity and
hydrodynamic force, which influence damage severity, varied significantly for different points at the same distance inland or
with similar inundation depths (Suppasri et al., 2013; De Risi et al., 2017). This was due to the differences in local topography
(Tsuji et al., 2014). Coastal topography influences tsunami behaviour on land, and therefore influences tsunami flow dynamics
and inundation characteristics (Suppasri et al., 2015). Previous studies have highlighted the importance of separating the two
types of coastlines when assessing tsunami damage (Suppasri et al., 2013; Tsuji et al., 2014; De Risi et al., 2017). This study
focuses on ports located in coastal plains, due to the (i) difficulty of accounting for complexity of flow processes in ria coasts
as well as (ii) significantly less port activity found in the narrow strips of ria coasts. Affected ports, namely Hachinohe, Kuji,
Ishinomaki, Sendai, Soma and Onahama, located in coastal plains were selected as study sites for our damage assessment (Fig.
103    1).

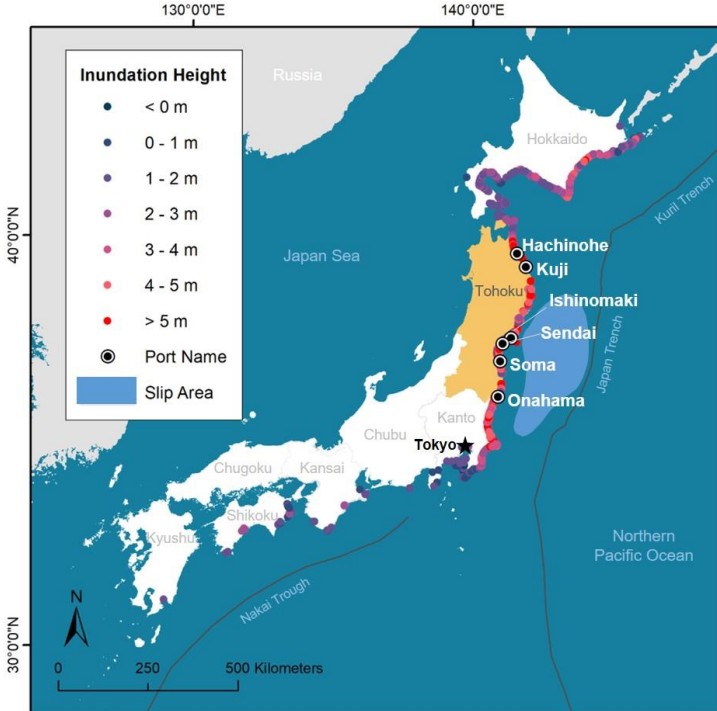

**Fig. 1.** Six of the affected ports (circled dots) were selected in this study due to similarities in their coastal morphologies –
they are located in coastal plains. Tsunami inundation heights were measured and collected by the Tohoku Tsunami Joint
Survey (TTJS, 2011) team. Inundation heights refer to the maximum height of tsunami inundation above the mean sea level
in Tokyo Bay (the Tokyo Peil datum). The generalized 2011 fault-rupture area (in light blue) was inferred from GPS data
adapted from Ozawa et al. (2011).



## 3. Workflow and data sources

A goal of this study was to produce tsunami damage fragility functions for industries commonly found in ports and their hinterlands, such as chemical and energy-related industries. The components required to derive fragility functions include the explanatory variable (hazard intensity measure), response variable (damage data) and a statistical linking model (Charvet et al., 2017). At present, a consolidated data source for tsunami damage to port structures has yet to exist. This data gap presents us with an opportunity to develop a damage database for port structures, and to use the damage data for the construction of fragility functions. We developed a framework (Fig. 2) for collecting and processing damage data within a database and using a machine learning workflow to evaluate those data and provide robust fragility functions; more details on our approaches are provided over the following subsections. We used freely available data where possible to illustrate how our methods can also be reproduced in other locations. A synopsis of the data used in this study and their sources are presented in Table 1.

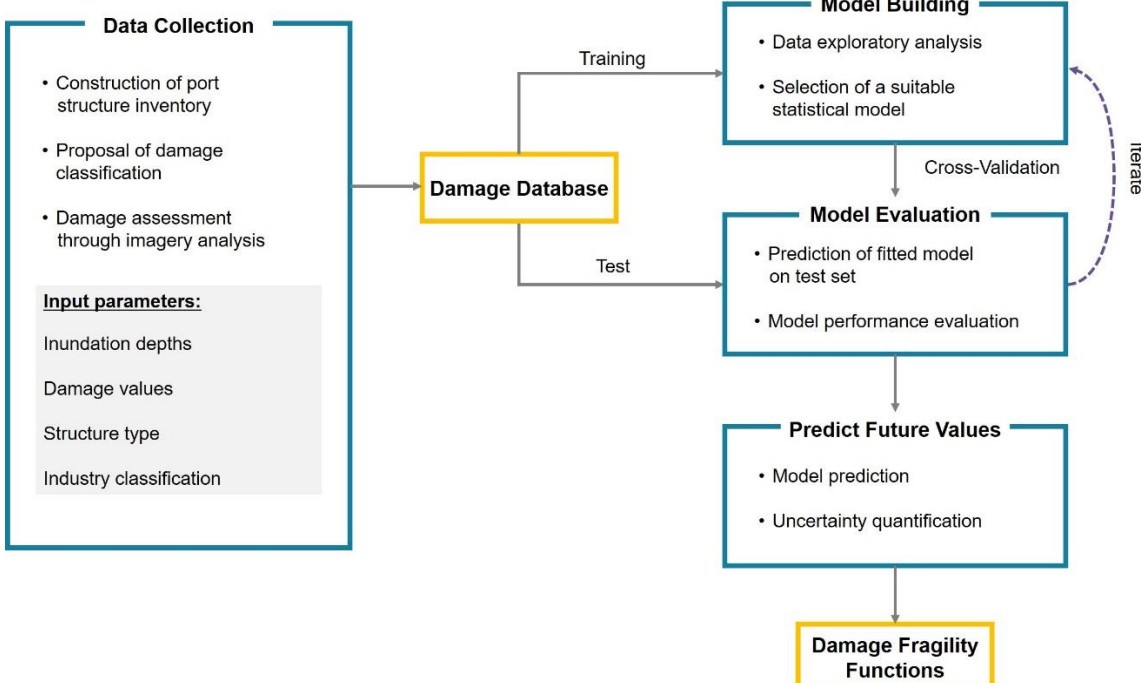

**Fig. 2.** The framework of this study follows the approach of a machine learning workflow. A damage database for port structures is constructed through data collection and processing. The consolidated data is then randomly split into training and test sets for model building and evaluation. This process is usually iterated until a satisfactory model is selected for the development of fragility functions. This is usually the case where are more than one model or parameter to choose from, whereas in our case, only inundation depth was considered as an explanatory variable.





129    **Table 1**. Data used in this study, their sources and the reference period from which data are taken.

| Data | Source | Data observation/ acquisition period | Citation |
|---|---|---|---|
| Tsunami inundation depths | Ministry of Land, Infrastructure, Transportation and Tourism (MLIT) | Mar 2011 – Dec 2012 | Ministry of Land, Infrastructure, Transportation and Tourism (2014) |
| Building database | Ministry of Land, Infrastructure, Transportation and Tourism (MLIT) | Mar 2011 – Dec 2012 | Ministry of Land, Infrastructure, Transportation and Tourism (2014) |
| Port structure footprint for digitisation | GSI Interactive Web: Map/Aerial Photo Browsing Service; | - | Geospatial Information Authority of Japan (2013) |
| | Google Earth engine | Mar 2009 – Sep 2010 | © Google Earth 2020 |
| Aerial images for damage assessment | Google Earth engine; | Mar 2009 – Sep 2010 + Mar 2011 – May 2011 ++ Feb 2012 +++ | © Google Earth 2020 |
| | GSI Map: Aerial Photo of Affected Area | Mar 2011 – May 2011 ++ Apr 2012 +++ | Geospatial Information Authority of Japan (2012a) |
| Oblique images for damage assessment | GSI Map: Oblique Photo of Affected Area | May 2011 ++ | Geospatial Information Authority of Japan (2012b) |
| Street view images for damage assessment | Google Street View | Jul 2011 – Aug 2011 ++ Aug 2013 +++ | © Google Street View 2020 |
| Landuse (industry) classification | Google Maps | - | © Google Maps 2020 |

+Pre-tsunami, ++Immediate phase after tsunami and  +++One to two years after tsunami (Intermediate phase) for damage assessment

130



## 4. Data collection

### 4.1 Establishing a damage database

The port structures referred to in this study collectively consist of a mixture of buildings and industry-related non-building structures (henceforth referred to as port infrastructure). Detailed building damage data have been collected by Ministry of Land, Infrastructure, Transportation and Tourism (MLIT, 2014) post-tsunami. However, the MLIT database predominantly consists of residential, commercial and some industrial buildings. Buildings within the port area are mostly missing from the database, and infrastructure such as silos, cranes and towers were not identified in the MLIT database.

To develop our own database of port structures, we extended the MLIT database, which already consisted of outlines of 3,057 buildings. To build the new database, port structure outlines (n = 2,173) were digitised into a Geographic Information System (ArcMap 10.5) using building footprints from the Geospatial Information Authority of Japan Interactive Map platform (GSI, 2013) as well as pre-tsunami aerial images from Google Earth Engine (Table 1). We identified 3,343 buildings and 1,887 infrastructure (5,230 total). The database is stored in the form of a Geographic Information System (GIS) attribute table. For each structure, we collected information on

    (1)   the type of industry

    (2)   the name of port

    (3)   the name of company at the time of tsunami (where available)

    (4)   maximum inundation depth values

    (5)   assigned damage state and,

    (6)   structure type (building or infrastructure)

### 4.2 Attributes of port structures and industry

Unique to this work, damaged structures were classified according to their industry type (Table 2). As with the construction of any fragility function, a key assumption is that structures under the same taxonomy are likely to perform similarly when exposed to a given hazard intensity (Pitilakis et al., 2014). For that reason, the classification of structures determines the robustness of the fragility functions developed. It was therefore important to create a suitable taxonomy for the types of structures being studied. Conventionally, building damage has been assessed by separating the buildings into their various construction types (e.g. masonry, wood, steel, unreinforced and reinforced concrete). Charvet et al. (2014) noted differences in the performance of buildings with different construction types to tsunami impacts following the Tohoku event. However, port structures consist of both buildings and infrastructure, with the infrastructure of a highly specialised nature where the design and construction criteria are industry-specific. A more suitable approach then would be to classify port structures according to their industry.





Different types of port activities occupy the port area. Aside from the core business of terminal operations, the port is also host
to distribution centres and non-maritime activities. To the best of our knowledge, there is no standard industrial classification
for port activities. We therefore proposed a broad classification for the port activities found in Tohoku ports, according to the
general industry that they fall into (Table 2). Classification for non-maritime port industries was adapted from the terminologies
used by European Sea Ports Organisation (ESPO, 2016) for the various industrial sectors found in European ports. We used
Google Maps and Google Street View to identify the business nature of each company (industry type), commonly through the
name of the company at the time of the tsunami. We identified eight main port industries based on our proposed taxonomy.

**Table 2.** Proposed classification for port activities found in the Tohoku region.

|  | Industry type | Description of port activities |
|---|---|---|
| Maritime industries | Cargo handling industry | Cargo handling services such as loading and unloading of ships (stevedoring) as well as the handling of cargo on shore. |
|  | Warehousing and distribution | Cold storage, warehousing and logistics support. |
| Non-maritime port-related industries | Chemical industry | Bulk chemical production e.g. alkane, propane and fertilisers. |
|  | Construction materials industry | Concrete and cement manufacturing. Asphalt and wood processing. |
|  | Energy-related industry | Coal power generation. Electric power generation and distribution. |
|  | Food industry | Seafood processing and food packaging. Feed manufacturing. |
|  | Manufacturing industry | Metal and alloy products. Plywood and paper products. |
|  | Petrochemical industry | Oil depots, reserves and refineries. |


### 4.3 Maximum inundation depths

Various tsunami hazard intensity measures (e.g. inundation depth, flow velocity and force) have been used in literature to
estimate structural fragility to tsunami impacts. Past studies (Macabaug et al., 2016; Park et al., 2017; Attary et al., 2019) have
shown that no single measure can fully characterise structural fragility to tsunami impacts as it is impossible to explain a
complex phenomenon through a sole parameter. For the purpose of this study, observed maximum inundation depth was
chosen as the representative intensity measure manifesting damage since depth is more easily estimated from field survey after
tsunami events as compared to other flow values, which typically have to be simulated. Using observational data also
minimises the uncertainty in intensity measure as compared to using simulated data (e.g. velocity and force).



Inundation characteristics were recorded and collected from a number of sources, namely tsunami trace heights by the Tohoku
Tsunami Joint Survey Group (TTJS, 2011), MLIT survey, photographs, videos, eyewitness accounts and other reports
(Leelawatt et al., 2014). The MLIT (2014) compiled all the maximum inundation depth values and building data into a single
database. Inundation depth refers to the depth of floodwater above ground. Each building surveyed in the MLIT database is
pegged with maximum inundation depth values, and where values were not available for some buildings (e.g. those that were
washed away), they were interpolated from nearby buildings with inundation depth values (De Risi et al., 2017). Similarly, for
buildings and infrastructure that were identified in this study, we interpolated inundation depth values from neighbouring
surveyed buildings.
**4.4 Proposed damage classification scheme**
For the first time, a damage classification scheme for tsunami damage to port structures is being proposed (Fig. 3). The MLIT
adopted a damage classification scheme for building damage assessment following the 2011 Tohoku tsunami (see Leelawatt
et al., 2014). Naturally, subsequent studies that used the MLIT damage database to analyse damage and derive fragility
functions followed the same classification scheme. The pitfalls of adopting the MLIT damage classification have been
highlighted in several studies (Leelawat et al., 2014; Charvet et al., 2015; Charvet et al., 2017). Firstly, the MLIT classification
consists of six damage states, which were found to have overlaps in their definitions (Leelawat et al., 2014; Charvet et al.,
2015). The overlapping definitions might have resulted in buildings being wrongly classified when performing damage
assessment. Ideally, damage states should be presented in a mutually exclusive and consecutive order (Charvet et al., 2015).
Secondly, descriptions in the MLIT classification scheme do not distinguish between structural and non-structural damage.
Therefore, the structural response of the buildings assessed is not being explicitly assessed. Additionally, by specifying the
range of inundation depths associated with each damage state, such definitions allude to inundation depths being a condition
of damage. This contradicts the objective of developing fragility functions as predictive models of damage. Over and above
the limitations outlined, the MLIT damage classification solely describes damage to buildings, which is otherwise unsuitable
for port structures.
To address the limitations of the existing damage classification of MLIT, we proposed a new damage classification for port
structures. This new classification scheme provides damage descriptions for both buildings and infrastructure. Degrees of
damage are classified into four levels (with damage state DS 0 being no damage), ensuring that the descriptions for each
damage state are mutually exclusive and in increasing order. Descriptions also include the expected serviceability of the
structure at each damage state. Pitalakis et al. (2014) argued that physical damages would reflect the expected serviceability
of the structure (condition for use) and its corresponding functionality (i.e. can its functions still be fulfilled?). The structural
integrity of port structures is also being considered. For instance, between DS 2 and DS 3, damage is distinguished by whether
it only affected non-structural components and/or roof (DS 2), or structural components such as columns and beams (DS 3).
We assumed that when the structural integrity of a structure is compromised, the structure would be removed.



| Damage State | Damage Description | |
|---|---|---|
| | Buildings (B) | Infrastructure (I) |
| DS 0 | • Little to no water penetration.<br>• Non-structural components (windows and door) and roof remain intact. | • No floodwater impacts on infrastructure. |
| | *Serviceability:* Ready for immediate use | *Serviceability:* Ready for immediate use |
| DS 1 | • Water penetration.<br>• Non-structural components and roof remain intact. | • No visible damage from outside of infrastructure. |
| | *Serviceability:* Ready for immediate use but requires interior restoration, such as drying of floors and walls, repainting, repairs to plumbing and electric systems. | *Serviceability:* Ready for immediate use. No obvious repair to infrastructure in the intermediate period after the tsunami. |
| DS 2 | • Non-structural components and/or roof have sustained damage.<br>• Structural components are intact. | • Some damage to infrastructure, while foundation or base remains intact. |
| | *Serviceability:* Obvious repair works in the intermediate period after the tsunami. Operational only after repairs. | *Serviceability:* Some form of repair to infrastructure in the intermediate period after the tsunami. Operational only after repairs. |
| DS 3 | • Structural components (columns and beams) have sustained damage, or rackings have buckled and folded. | • Foundation or base of infrastructure has folded or buckled. |
| | *Serviceability:* Not repairable. Replacement or removal of building in the intermediate period after the tsunami. | *Serviceability:* Not repairable. Removal or replacement of infrastructure in the intermediate period after the tsunami. |
| DS 4 | • Total structural failure.<br>• Building has either overturned or slid from original position. | • Infrastructure has overturned or slid from original position. |
| | *Serviceability:* Not operational. | *Serviceability:* Not operational. |

**Fig. 3.** Proposed new damage classification for port industries. Descriptions for damage to both buildings and non-building infrastructure are provided in the classification table. DS 1 and DS 2 are considered as non-structural damages, while DS 3 and DS 4 are structural damages.

**4.5 Damage assessment through spatio-temporal analysis**

A combination of free-to-use sources were used to inform our classification decisions when assigning damage states to individual port structures (Table 1). Port structures were assessed through the analysis of satellite imagery, using pre- and post-tsunami images from Google Earth engine and Geospatial Information Authority (2012a), as well as photographic interpretations of post-tsunami oblique images from Geospatial Information Authority (2012b). Pre- and post-tsunami images refer to observations made before 11 March 2011, and on and after 11 March 2011 respectively (Table 1). Apart from aerial



222 and oblique images, we visually assessed the conditions of port structures through Google Street View images. Google Street

223 View, a service available on Google Maps web, provides panoramic view of the landscape at a street level. An example of

224 how a building or infrastructure was being assessed is illustrated in Fig. 4.

225 The three types of images (aerial, oblique and street view) provided different, yet complimentary, types of information. Aerial

226 images were particularly useful in assessing washed away and collapsed structures (DS 4). Street View images were used to

227 identify damage from façade level, which supplemented as "ground truth surveys". The high-resolution imagery provided by

228 Google Street View allowed us to pick up finer details such as structural and non-structural damage to port structures, which

229 would otherwise be missing from aerial imagery. However, because Street View imagery was captured through vehicle-

230 mounted cameras, the availability of these images are constrained by the accessibility of roads by the vehicle at the time of

231 survey. Where imagery was not captured by Google Street View due to such constraints, we capitalised on the alternative

232 views provided by GSI oblique images.

233 Advances in mapping technologies mean that temporal changes are also being captured and documented in these mapping

234 applications. The time-slider functions on Google Earth engine and Google Street View web, as well as the date stamps on

235 GSI images, allowed us to review temporal changes in the built environment. For images in Google Earth and Google Street

236 View, different phases of the tsunami, i.e. pre-tsunami (before March 2011), immediately after the tsunami (up to 6 months

237 after the tsunami) and the intermediate recovery phase (1 – 2 years), were all captured in the same point locations. With

238 coordinates being embedded in the aforementioned data sources, we were also able to reference GSI aerial and oblique post-

239 tsunami images to the same locations. The large amount of high-quality data provided by these image databases and mapping

240 applications have been a large driver of our data collection in this study.



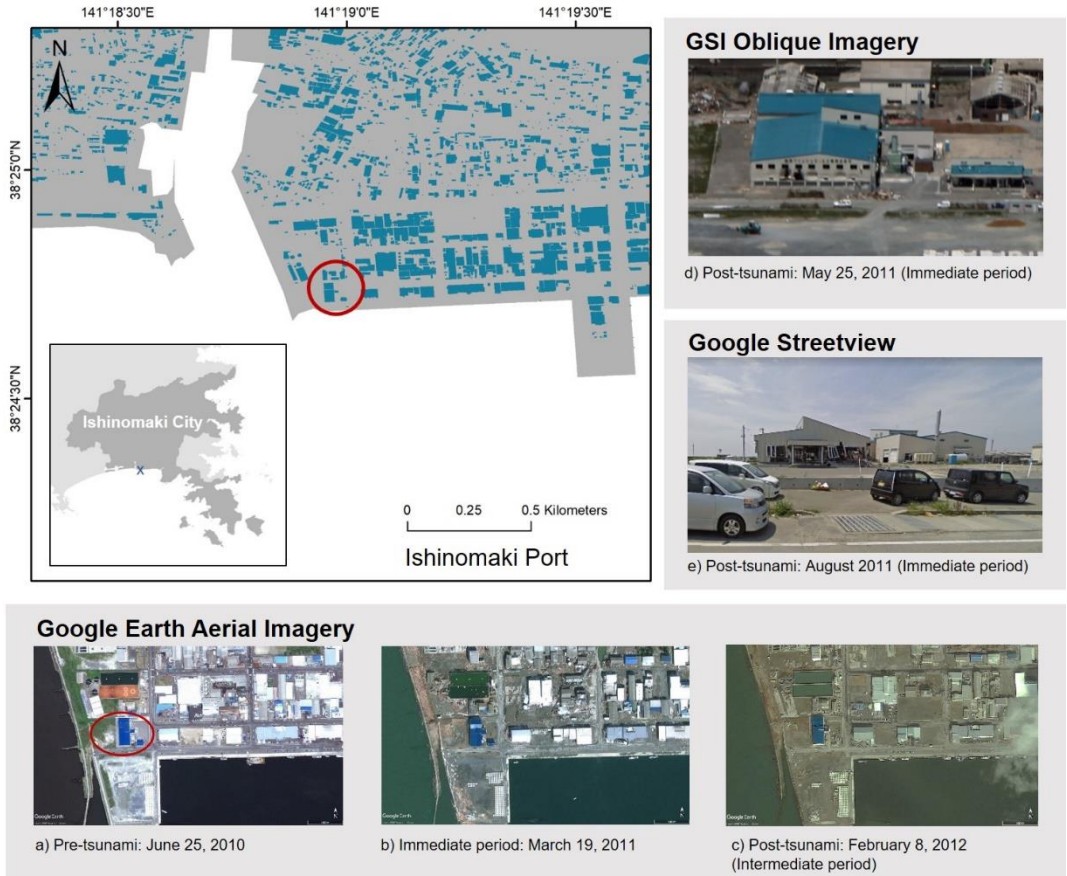

**Fig. 4.** A building (circled in red) in Ishinomaki Port has been selected to demonstrate how spatiotemporal damage assessment had been conducted in this study. For every port structure, we reviewed four main sources of data (©Google Earth 2020, ©Google Street View 2020, GSI Aerial and Oblique images) to estimate the level of damage sustained during the tsunami.

## 5. Model building

Fragility functions describe the probabilities of damage exceedance for a given intensity measure or flow characteristic. The probability of damage exceedance can simply be expressed as:

$$PDS = P(ds \geq DS \,|\, IM)$$

, where $ds$ is the observed damage state of a structure, $DS$ the classification provided by the damage scale and $IM$ the intensity measure (Charvet et al., 2017). In the case of this study, tsunami inundation depth was used as an explanatory variable in the prediction of structural damage probability. Typically, empirical tsunami fragility functions are constructed by fitting an appropriate statistical model to post-tsunami damage data.





## 5.1 Evaluation of statistical models available

In recent years, a number of studies evaluated the suitability of various statistical models in representing tsunami damage to structures (Charvet et al., 2014; Macabaug et al., 2016; Charvet et al., 2017). Parametric (e.g. Ordinary Least Square regression, Generalised Linear Model or ordinal logistic regression models), semi-parametric (e.g. Generalised Additive Model) and non-parametric (e.g. Kernel Smoother) statistical model types are amongst the most commonly used. These statistical models are extensively reviewed in Rosetto et al. (2014), Lallemant et al. (2015), Macabaug et al. (2016) and Charvet at al. (2017), and readers are referred to these studies for a more comprehensive understanding of the advantages and disadvantages of using the various types of statistical models.

Generalised Linear Models (GLM), an extension of classical linear regression models, have been recommended as more reliable forms of fragility functions for the following reasons:

- Discrete probability distributions can be used to predict discrete responses (Charvet et al., 2017). This is especially important for categorical data (such as damage states), because it is statistically incorrect to assume that the difference between categories is linear/continuous, e.g. the difference between DS 1 and DS 2holds the same meaning for the difference between DS 2 and DS 3 (Guisan and Harrell, 2000).

- Unlike classical linear regression models (e.g. ordinary least square regression) which assume either a normal or lognormal distribution, the response variable need not be normally distributed and can take on any of the exponential family distributions.

- It does not assume a linear relationship between the explanatory variable and response variable, but a linear relationship is assumed between the transformed response through a link function and the explanatory variables.

- Maximum likelihood estimation (MLE) is used rather than ordinary least squares to estimate the parameters. MLE has the advantage of explicitly reflecting the probability distribution of the random variable of interest.

- Overfitting of data can be avoided by using cross-validation analysis to determine optimal model parameter values.

- Model uncertainty can be quantified by supplementing the median of the response with confidence or prediction intervals.

## 5.2 Data exploratory analysis

The response variable is ordinal (in the sense that DS 0 < DS 1 < DS 2 < DS 3 < DS 4). A visual inspection of the distribution of depth given damage data (Fig. 5) indicates non-normality, with the distribution skewed towards the right, indicating a lognormal transformation of inundation depth variable would be appropriate. Frequency counts of the damage data show that damage state (DS 1) makes up the majority of the dataset (n = 2710), and DS 3 and 4 a much smaller proportion (n = 576 and n = 605 respectively).

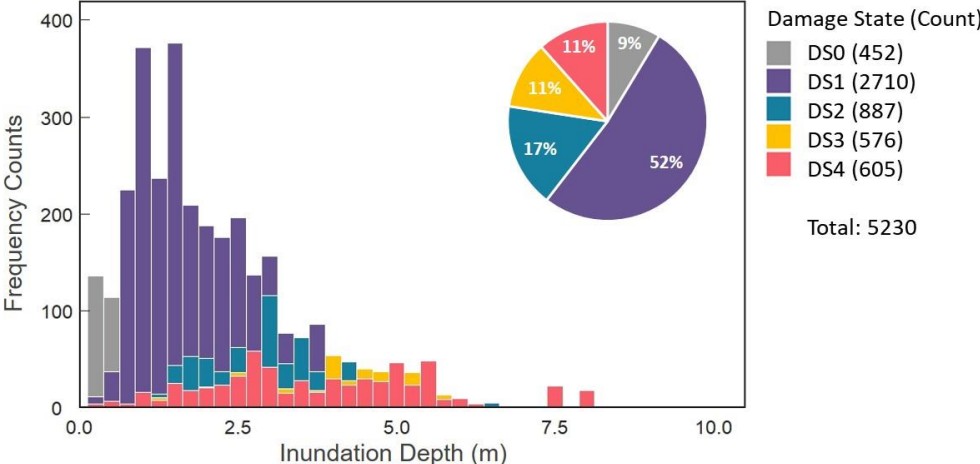

283

**Fig. 5.** Histograms of each damage state. Distribution of damage data indicates non-normality and DS 1 accounts for the majority of the dataset. Outliers exist in DS 3 and 4, with no damage states recorded for inundation values between 6 to 7.4 metres. Outliers are not removed from the model, as they are legitimate observations and possible outcomes.

**5.3 Selection of a suitable statistical model**

An ordinal logistic regression model, an ordinal and logistic recourse of GLMs, is adopted. It has the additional advantage of accounting for and maintaining the ordered nature of damage-state data. As this model recognises the ordered nature of the damage states, overlapping pathways of the fragility functions can be avoided (Charvet et al., 2017). Overlapping fragility functions, as is common when fitting separate GLMs, may unwittingly imply that the probability of a higher damage state (e.g. DS 4) being exceeded is higher than that of a lower damage state (e.g. DS 3) as inundation depth increases. Ordinal models also make full use of the ranked data rather than simplifying it into binary exceedance and non-exceedance, and therefore preventing the loss of information (Ananth and Kleinbaum, 1997).

The dependence of the response variable DS on predictor variable X can then be represented as follows

$$P_{DS} = P(ds \geq DS_i | X_j)$$

, where $DS_i$ refers to the $i_{th}$ damage state, $j$ the specified predictor (IM) or combination of predictors. The model relates the probability of the outcome, $P_{DS}$, to all explanatory variables $(X_1, X_2, .., X_j)$ through a linear predictor. There are three basic components to any GLM, and Table 3 describes the components in the context of the ordinal logistic model used in this study.



**Table 3.** Components of an ordinal logistic regression model

| Random Component | *The probability distribution of the response variable.* |
|---|---|
| | A multinomial distribution is assumed for the cumulative probabilities in an ordinal logistic regression model. |
| Systematic Component | *The explanatory variable $(X_j)$ or the linear combination of the explanatory variables $(X_1, X_2, .. , X_j)$ in creating the linear predictor e.g. $\beta_0 + \beta_1 X_1, \beta_2 X_2 + \cdots + \beta_j X_j$ , where $\beta_0$ and $\beta_{1,j}$ are transformed constant and regression coefficients through maximum likelihood estimation.* |
| Link function | *The link between random and systematic components.* |
| | Describes how the cumulative probability $P_{DS_i}$ of the expected outcome for any damage state $DS_i$ relates to the linear predictor of explanatory variables $X_j$. In this instance, the link function chosen takes on a logit form g where $$g\left(P_{DS_i}\right) = \log\left(\frac{P_{DS_i}}{1 - P_{DS_i}}\right)$$ , with $$P_{DS_i} = P\left(ds \geq DS_i \middle| X_j\right) \quad \forall\, i \in (1, \dots, I)$$ Therefore, the dependence of the response variable DS on the linear predictor can be re-expressed as $$\log\left(\frac{P_{DS_i}}{1 - P_{DS_i}}\right) = \beta_{0,i} + \beta_1 X_1 + \beta_2 X_2 + \cdots + \beta_J X_J$$ $$\log\left(\frac{P_{DS_i}}{1 - P_{DS_i}}\right) = \beta_{0,i} + \sum_{j=1}^{J} \beta_j X_j$$ The corresponding regression coefficients $\beta_{1,j}$ in the link function are fixed across every damage state except for the intercept, so as to maintain the order of the response categories. |






The conditional probability $P(ds \geq DS_i|X_j)$ is a common vector of regression coefficients β, which connects probabilities for
varying levels of damage. When expressing the cumulative probabilities of each damage state as separate curves, the
relationships between damage states in increasing order of severity are defined as follows:
$$P_{DS} = P(ds = DS_i \,|IM = X_j) = \begin{cases} 1 - P(ds \geq DS_i|X_j) & i = 0 \\ P(ds \geq DS_i|X_j) - P(ds \geq DS_{i+1}|X_j) & 0 \leq i \leq N_{DS} \\ P(ds \geq DS_i|X_j) & i = N_{DS} \end{cases}$$


, where $N_{DS}$ refers to the number of damage states, including DS 0 (Macabaug et al., 2016).

## 6. Model evaluation

### 6.1. 10-fold cross-validation

Model accuracy was used as a quantitative indicator of the performance of our models. We wanted to assess the goodness-of-
fit of the models and determine its predictive ability. It was difficult to test the predictive ability of our models where there
were no further samples to test with. In order to optimise model design while preventing overfitting, the cross-validation
method was applied to evaluate the prediction accuracy of our models. Cross-validation techniques make use of the available
dataset by dividing them into two subsamples – one to train the model and the other to predict the model on.
One cross-validation technique is K-fold, where the dataset is divided into K number of approximately equal-sized subsets as
illustrated in Fig. 6a. One subset is taken out as a test set for validation, and the remaining K – 1 subsets are then used to train
a model. This hold-out method is then repeated for K number of times, with a new subset being used as a test set in each
iteration. Only after all K models are fitted, statistics of the model performance are tabulated. For the purpose of this study, a
10-fold cross-validation approach was taken.
The accuracy of a model is determined by the proportion of correctly classified responses. When applied to the k-fold
technique, the fitted model is used to predict response on the held-out k[th] subset in each iteration. The recorded response is
tabulated against actual observations in the k[th] subset and a confusion matrix is constructed as demonstrated in Fig. 6b. The
diagonal of the confusion matrix represents the sum of correctly predicted response, the proportion of correctly classified
response is then calculated by
$$Accuracy = \frac{Sum\ of\ correctly\ predicted\ response}{Sum\ of\ total\ observations}$$


Accuracies are recorded in each iteration of the K-fold, and the mean and standard-deviation of the tabulated accuracies are
taken to assess the predictive ability of the model. All statistical analyses and modelling in this study were carried out using
the statistical software R (R Core Team, 2020).



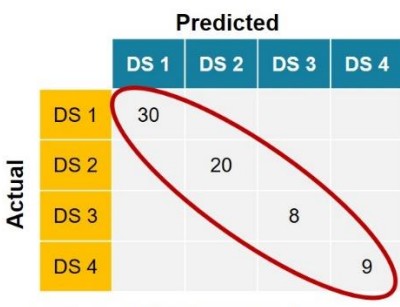

**Fig. 6.** (a) An example of a 5-fold cross-validation technique for the purpose of illustration. The same dataset can be folded into 5 equal sizes, and one fold is held-out for testing and the remaining 4 folds are used to develop a training model to predict the accuracy of the training model. This is repeated 5 times, with accuracies being tabulated in each iteration. (b) Accuracy table (confusion matrix) is produced in each iteration of the k-folds. The sum of the diagonal in the table is divided by the sum of observations to get the percentage of accuracy in the $k^{th}$ fold.

## 6.2 Quantification of uncertainty

The fragility functions, when presented as curves or plots, represent the expected value of the response variable. Therefore, they represent only a sample estimate of the population values. Statistical variations of the fragility functions can be accounted for by estimating the confidence intervals. In this study, we adopted bootstrap-based confidence intervals to estimate the uncertainty in estimation or prediction. The bootstrap method treats the original dataset of values as a realised sample from the true population and does not make any assumptions about the underlying distribution of the population parameters (Yung and Bentler, 1996). Values from the original dataset are resampled repeatedly, with replacement. This was done for 1000 iterations, with the predicted logit computed in each iteration. To derive a 95% confidence band, the $2.5^{th}$ and $97.5^{th}$ quantiles of the 1000 estimates were drawn at each inundation depth interval (0.01m).

## 7. Results

### 7.1. Damage database for port structures

To characterise the vulnerability of assets in various port industries, damage assessment was performed for buildings and infrastructure in the Tohoku region. We compiled damage information on port structures into a database, which is available online through an unrestricted data repository (DR-NTU) hosted by Nanyang Technological University (https://doi.org/10.21979/N9/OTZMT1) (Chua et al., 2020).

The port damage database consists of 5,230 port structures, of which 3,343 are buildings and 1887 are infrastructure. The port structures were identified in six case study ports, across eight port industries. The damage dataset show that most port structures



sustained minimal structural damage classified as damage state DS 1 (Table 4). Consistently for all port industries, the majority
of the observed damage corresponds to DS 1 (Fig. 7.) Notably, many industries such as chemical, petrochemical and energy-
related industries sustained minimal structural damage mainly due to flooding at DS 1, which only required some clean up and
interior restoration and remained mostly operational after restoration. On the other hand, cargo handling and food industries
sustained a wide range of damage from minimal damage (DS 1) to total damage (DS 4), corresponding to nearly all damage
states. Tsunami floodwaters at depths of less than 5 metres inundated most port structures. In extreme cases, inundation depths
affecting port structures reached as high as 7.5 metres. The minimum recorded inundation depth was 0.1 m.

**Table 4.** Summary of port structures identified in the various ports, sorted according to their industries.

| | North Tohoku | | South Tohoku | | | | |
| --- | --- | --- | --- | --- | --- | --- | --- |
| | **Hachinohe** | **Kuji** | **Ishinomaki** | **Sendai** | **Soma** | **Onahama** | **Total** |
| Cargo Handling Industry | 31 | 9 | 31 | 32 | 25 | 62 | 190 |
| Warehousing and Distribution | 111 | 16 | 175 | 105 | 39 | 17 | 463 |
| Chemical Industry | 236 | - | 208 | 27 | 85 | - | 556 |
| Construction Materials Industry | 29 | 20 | 20 | 99 | 9 | 37 | 214 |
| Energy-related Industry | 125 | - | - | 104 | 134 | 50 | 413 |
| Food Industry | 12 | 37 | 430 | 151 | - | - | 630 |
| Manufacturing Industry | 1010 | 60 | 587 | 279 | 144 | - | 2080 |
| Petrochemical Industry | 202 | 41 | 38 | 324 | - | 79 | 684 |
| **Total** | | | | | | | 5230 |





**Fig. 7.** Data attributes of the port industries affected by the 2011 Great East Japan tsunami.



## 7.2 Damage fragility functions for port industries

Damage fragility functions were produced for eight major port industries as depicted in Fig. 8. Individual fragility curves were plotted for each damage state and the solid lines represent the probabilities of a structure exceeding each damage state given a range of inundation depths and the shaded regions their corresponding 95% confidence intervals.

The fragility functions (Fig. 8) suggest that chemical, cargo handling, and construction materials industries are more vulnerable. Higher probabilities of damage exceedance are reached at a more rapid rate as compared to other industries. In contrast, energy-related industry and warehousing and distribution are showing a gentler incline in damage probability for higher levels of damage (DS 3 and DS 4), indicating a greater resistance to tsunami impacts. A key assumption of fragility studies and of this study is that damage is directly related to the properties of the elements at risk. Thus, aside from tsunami intensity measures, the composition and structural design of each industry could determine the differences in vulnerabilities. For example, power plants (energy-related industries) and warehouses are structurally robust by design. Most heavy equipment found in power plants is normally supported in large reinforced concrete foundations or housed in large steel structure buildings (Cruz and Valdivia, 2011) and is therefore more resistant to tsunami loads. Likewise, many warehouses in the studied ports were reinforced concrete buildings with their warehouse floor raised above road levels, which increased the height of non-structural elements (e.g. docks and doors) relative to tsunami inundation. Comparatively, chemical facilities typically consist of more fragile components which are not part of the primary load resisting systems such as pipelines, pumps, compressors and tanks, and they are extremely vulnerable to damage from tsunami inundation and forces. As observed in the 2011 event, hydrodynamic and hydrostatic forces from the tsunami resulted in the breaking of pipe connections, floating tanks and overturning of unanchored infrastructure (Krausmann and Cruz, 2013). Meanwhile in cargo handling facilities, loading and unloading infrastructure was mostly anchored, but instances of cracked pavements and damaged crane rail foundations by the earthquake and tsunami were reported to result in the derailment and collapse of cranes (Technical Council on Lifeline Earthquake Engineering, 2017). Nonetheless, other factors such as debris impact and proximity to shoreline should not be discounted when considering the differences in the response of each industry to tsunami impacts.

For each damage state, we considered the minimum depths where damage exceedance probability reaches near 1 or becomes nearly certain. Minimum damage (DS 1) is almost certain at 2.5 m consistently for all industries except energy-related industry. DS 1 occurs when there is water penetration into the building and interior restoration is required (Fig. 3). Logically, water penetration into buildings would be expected from 0.45 m since buildings are required to be constructed 0.45 m above road level as specified by the Building Standard Law of Japan (Building Centre of Japan, 2013). Threshold depths for DS 1 might have occurred at 2.5 m because of the aggregation of data for both infrastructure and buildings. We observed that there were many buildings (especially warehouse) and infrastructure such as storage tanks and silos that were elevated above ground and therefore, the number of exposed assets at lower inundation depths were reduced. The trend for other damage states is however not obvious and it is difficult to pinpoint minimum depth values where damage becomes certain.





A threshold value is said to be reached when damage curves from all states of damage converge at the probability of near
100%. Key threshold value can be defined as the parameter (in this case, inundation depth) criteria at which DS 4 (collapse)
becomes certain. Earlier studies of the 2011 Great East Japan tsunami (Suppasri et al., 2013; Charvet et al., 2014) examined
the key threshold values for buildings, using damage data provided by MLIT. Suppasri et al. (2013) identified 2 m to be the
key threshold value for all building types. More recent analysis found inundation depth thresholds to differ between
construction types: from 2 m for wooden buildings (Charvet et al., 2014) to more than 10 m (Charvet et al., 2015) for steel and
reinforced concrete construction types. Similar patterns have emerged in the present analysis. The near 100% probability of
collapse occur at inundation depth exceeding 10 m for all industries. As such we were unable to quantify the key threshold
values for collapse for port industries. There are several possible reasons for this observation. Two likely explanations stand
out. The first being port structures are structurally much more resistant to tsunami loads than regular low-rise buildings because
industrial buildings and structures are designed to withstand greater loads, including but not limited to dead loads, live loads,
wind and earthquake loads. Therefore, greater tsunami inundation depths are required to overcome the resistance of port
structures. A second possible explanation is that inundation depth alone is insufficient to explain damage, although it provides
a first indication.
The effects of uncertainty were quantified through the construction of confidence intervals around the mean of the resulting
probabilities. Confidence intervals around DS 1 are consistently narrow in width for all industries (Fig. 8), which could be
associated with its large sample size. Contrastingly, for higher levels of damage (DS 3 and DS 4), confidence intervals tend to
widen towards higher inundation depths. An observation made in the process of damage data collection through photographic
interpretations was that many structures sustained very little damage despite high inundation depth values, which explains the
smaller sample sizes and therefore wider confidence intervals for DS 3 and DS 4 at higher depth values. In the same way,
industries with the widest confidence intervals such as cargo handling industry and construction materials industry tend to
have smaller sample sizes. By contrast, variabilities around the median curves tend to be smaller for the manufacturing
industry, food industry, warehousing and distribution and petrochemical industry due to their larger sample sizes.

**Fig. 8**. Fragility curves with 95% confidence bands for port industries identified in this study. Chemical, cargo handling and construction materials industries appear to be more vulnerable to tsunami inundation depths, while petrochemical and warehousing and distribution industries have lower damage probabilities for the same inundation depths. Wider confidence bands imply greater variability in uncertainty and could be results of smaller sample sizes.





## 8. Discussion

### 8.1 Comparison of damage dataset with functionality of port industries post-tsunami



We compared the damage database with existing literature to validate our observations. Most of the existing literature are
either limited to descriptive analysis of damage to port facilities or are not available in English. We found only one study to
be comparable with this study, in terms of the quantification of damage to port industries. A post-2011 tsunami survey was
carried out by the Tohoku Regional Development Bureau (MLIT, 2011) between October and November, 2011. We considered
the survey period as the intermediate period for reconstruction after the tsunami. The survey is a questionnaire survey on the
recovery status of companies in tsunami-affected ports, including ports outside of our study sites. 226 of the 233 companies
found in the affected ports responded to the survey. Findings from the survey were adapted from MLIT (2011) and we have
translated them into English (Fig. 9).
We drew comparisons between the recovery status of the companies affected (MLIT survey) and the serviceability of port
structures at each damage state (this study). It is difficult to make a direct comparison between the two. While port structures
are the physical components of these companies, port structures and companies are inherently different entities. Therefore, an
assumption made here is that the serviceability of port industries is indicative of the recovery status of the companies surveyed
in the MLIT survey.
13% of the companies were found to be unaffected by the tsunami (Fig. 9), which marks a good agreement with our study
where port structures sustaining no damage (DS 0) makes up 9% of the dataset (Fig. 4). In addition, approximately 12% of the
companies found to be unrecoverable, which we assume to correspond to damage state DS 4 (11%) in our study. The MLIT
survey found 72% of the companies to be in various stages of recovery during the survey and a majority (46.8%) of the
companies were almost fully recovered (> 80% recovery) in the intermediate phase. Similarly, a large proportion (52%) of our
damage data falls into DS 1 where port structures can be operational almost immediately after tsunami (Fig. 3). It is
challenging, however, to draw parallel between the degrees of recovery with the damage states presented in this study. We
stress that this approach is a relative measure of the validity of our dataset and damage assessment. Nonetheless, we can infer
that damage observations made from photographic interpretations in this study are rather similar to actual observations.

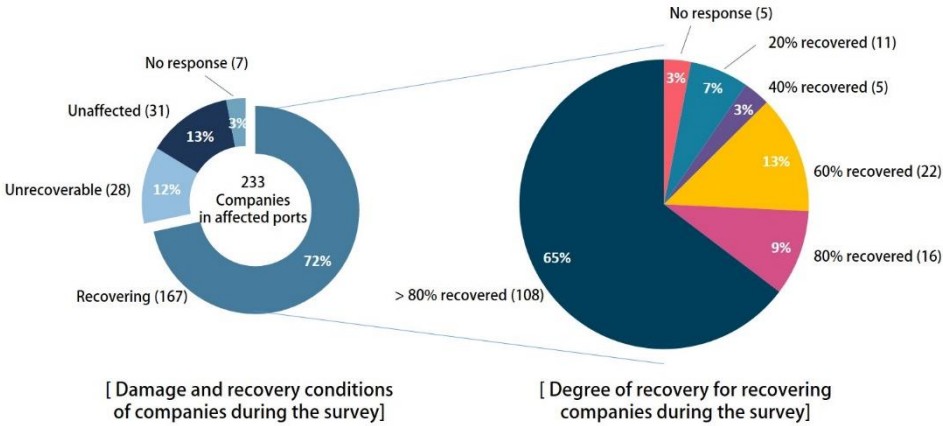

[ Damage and recovery conditions
of companies during the survey]

[ Degree of recovery for recovering
companies during the survey]


**Fig. 9**. Damage conditions and degrees of recovery of companies in the tsunami-affected ports of Hachinohe, Kuji, Miyako,
Kaimaishi, Ofunato, Ishinomaki, Sendai-Shiogama, Soma and Onahama. 65% of the recovering companies were almost close
to full recovery (>80%) at the time of the survey. Adapted and translated from MLIT (2011).
**8.2 Fragility models and their classification accuracies**
Using the 10-fold cross validation technique, we evaluated the prediction accuracies of our models. Mean accuracies and their
standard deviations for each industry are illustrated in Table 5. Port structures have an overall accuracy of 59%. The
petrochemical industry, energy-related industry, chemical industry and manufacturing industry display higher accuracies –
75%, 70%, 69% and 64% respectively. In contrast, warehousing and distribution industry, cargo handling industry and food
industry display lower prediction accuracies – 40%, 38% and 28% respectively.
We looked at the underlying nature of our datasets to better understand the differences in accuracies. The petrochemical
industry, energy-related industry, chemical industry and manufacturing industry display higher accuracies and are represented
by large sample sizes (Fig. 7). On the contrary, the cargo handling industry is represented by only 190 data points. However,
because the food industry is represented by a large sample size but seemingly displays very low accuracy, we were unable to
conclude that sample size has an influence on the accuracies of the fragility models. In addition, the three industries
(warehousing and distribution, cargo handling and food industries) which display low accuracies are well represented across
all damage states.
The intrinsic differences between industries could have an effect on reducing accuracies. The composition of buildings and
infrastructure differ from industries to industries. For instance, cargo handling industry, which displays lower accuracy,
typically consists of mobile equipment such as cranes and conveyors as well as temporary transitional storage and components
such as chillers and tanks. Damage to transient port structures as such may be reflected in the damage data as part of the overall
assessment and introduce noise to the damage data, thus reducing model accuracy. In addition, the structural design of port
structures may vary between facilities of the same industry. For example, warehouses in the studied ports were mostly
reinforced concrete buildings, but some were made of mixed materials such as reinforced concrete foundations with light metal





or masonry walls. Whereas power plants (energy-related industry) and petrochemical industry are consistent in construction
material and more robust by design, which perhaps explain their higher accuracies. Thus, variability between port structures
of the same industries can also impact accuracy if those variables are not accounted for in the models.
Another possible explanation is that many assets might have sustained extensive damage from earthquake activities such as
ground motion and liquefaction prior to the tsunami, as was observed by Kazama and Noda (2012). A preliminary inspection
of the damage dataset indicated a greater representation of data from ports that have experienced stronger ground motion for
the following industries – food, cargo handling and warehousing and distribution (Table 4). On the other hand, industries that
display higher accuracies have a greater data representation from ports that were not as severely affected by ground motion.
The significance of this relationship between the effects of the preceding earthquake and the damage observed is further
investigated in the proceeding section.
For most industries, our models performed better in terms of their classification accuracies as compared to fragility models
developed for buildings using the MLIT damage classification, which were found to have an accuracy of 52% (Leelawat et
al., 2014). As this is the first time tsunami damage is being quantified as a response of inundation depth for port industries, we
have no other models that we could use for comparison.
**Table 5.** Mean accuracies and standard deviations of accuracies of the various port industries.

| Industry Type | Mean Accuracy | SD Accuracy |
|---|---|---|
| Cargo Handling Industry | 0.374 | 0.221 |
| Warehousing and Distribution | 0.397 | 0.198 |
| Chemical Industry | 0.687 | 0.300 |
| Construction Materials Industry | 0.502 | 0.285 |
| Energy-related Industry | 0.707 | 0.245 |
| Food Industry | 0.283 | 0.204 |
| Manufacturing Industry | 0.638 | 0.249 |
| Petrochemical Industry | 0.746 | 0.218 |
| All Industries (Whole Tohoku) | 0.587 | 0.203 |




### 8.3 Effects of pre-tsunami earthquake activities on observed damage to port structures

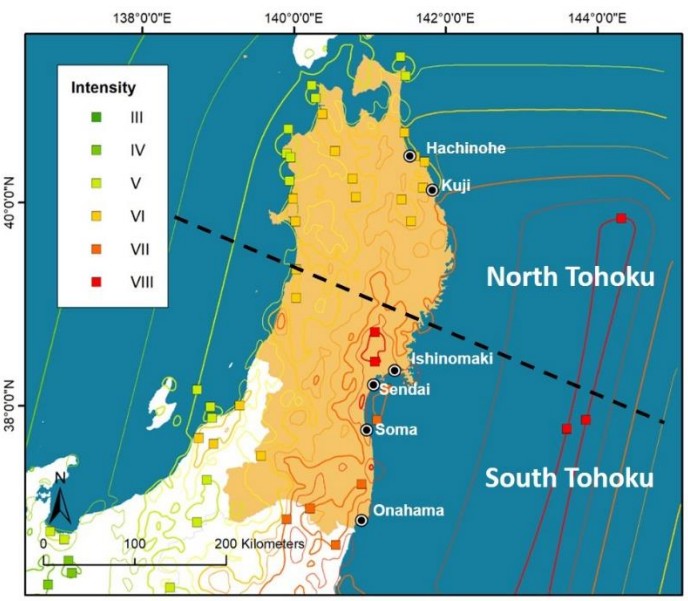

**Fig. 10**. Mercalli intensities (MI) recorded by United States Geological Survey (USGS, 2020) for the Great East Japan earthquake and tsunami. Earthquake intensities differ between the northern (MI VI) and southern (MI VII - VIII) regions of Tohoku. North Tohoku experience less effects from ground shaking than in the South.

One of the concerns raised in the process of this research was the effect of ground motion, which preceded the arrival of the tsunami, on asset damage. The effect of ground motion on damage to coastal structures was studied by Sugano et al. (2014). The authors noted that in the northern Tohoku region, only little damage was sustained due to ground motion and the damage observed was to a greater effect due to tsunami inundation. On the other hand, damage due to ground motion was substantially greater in southern Tohoku region, more specifically coastal areas south of Miyagi Prefecture. Similar observations were made by Okazaki et al. (2013), whom conducted surveys in Ishinomaki and Sendai ports and found that the two sites were exposed to both severe ground motions and great tsunami wave heights. Kazama and Noda (2012) have also highlighted the possibilities of liquefaction prior to the arrival of the tsunami but noted the impossibility of identifying locations of which liquefaction had occurred after the tsunami.

To assess if ground motion-induced damage affects the accuracies of our models, we separated the damage data according to the locations of ports (between northern Tohoku and southern Tohoku regions). The ports of Hachinohe and Kuji fall within the northern region, and the ports of Ishinomaki, Sendai, Soma and Onahama are located within the southern region (Fig. 10). We selected two industries to capture the effect of ground motion, instead of using the entire dataset since it has the effect of aggregating data from different industries and hence neglect differences in their physical characteristics. The manufacturing industry was considered because of its high prediction accuracy and its large sample size. The food industry was also





considered due to its poor prediction accuracy – we wanted to examine if pre-earthquake activities might explain the poor
prediction ability of the fitted model.
Damage data for both industries was split into two sites (North and South Tohoku). For each dataset, an ordinal regression
model was fitted and its response was captured in a 10-fold cross-validation. The resulting fragility models and their mean
accuracies are shown in Fig. 11. We observe that port structures in South Tohoku tend to reach high probabilities of non-
structural (DS 1 and DS 2) damage at lower inundation depths than structures in North Tohoku. This suggests that earthquake
damage might have weakened structures prior to the tsunami, leading to a steeper incline in damage probabilities as compared
to structures in North Tohoku. However, at higher levels of damage (DS 3 and DS 4), ground shaking appears to have had less
influence on damage. For both industries in the northern region, models depict a smaller initial increase in damage for higher
levels of damage DS 3 and DS 4 but probabilities incline more rapidly at higher inundation depths. The opposite holds true for
both industries in the southern region, i.e. damage probability for DS 3 and DS 4 incline at a slower rate at higher inundation
depths implying that a larger depth is required to induce structural damage (DS 3) and collapse (DS 4). Ground shaking
therefore only influenced lower levels of damage, tsunami inundation and flow characteristics still had a greater influence on
higher levels of damage.
The mean accuracies of using only datasets from North Tohoku are significantly higher than those of South Tohoku datasets.
It appears that the aggregation of datasets from the two environments has the effect of averaging the mean accuracies for the
whole region (Table 5, Fig. 11). It suggests that damage sustained by port structures in the Southern Tohoku region was
influenced by the compound effects of earthquake and tsunami loads. Inundation depth alone is not sufficient to explain the
damage observed. However, as Charvet et al. (2014) pointed out, it is difficult to distinguish the extent to which buildings had
already been affected by earthquake damage prior to the arrival of the tsunami. Therefore, it was difficult to separate the effects
of ground motion and liquefaction when we developed our fragility models.
There are other factors such as debris impact, the effect of shielding and local characteristics of the built environment that may
have influenced the results observed (Tarbotton et al., 2015). Regardless, we note that while the fragility model developed for
food industry using only data from the North has an improved mean accuracy, there is a substantial increase in the uncertainty
of the model (Fig. 11). It is not surprising as wider confidence intervals are a reflection of a limited sample size. An unbiased
sample is not representative of the whole population, and therefore, it is prudent that all available samples are used to fit the
fragility functions.

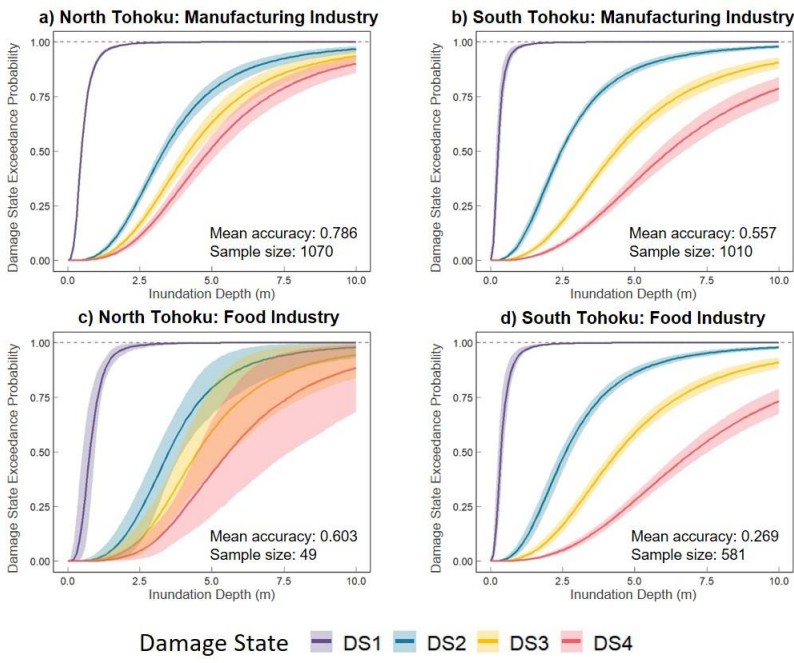


**Fig. 11.** Fragility functions developed for manufacturing industry in (a) North Tohoku, (b) South Tohoku as well as food industry in (c) North Tohoku and (d) South Tohoku. To evaluate the effects of preceding earthquake damage on overall damage assessment, datasets for each industry were divided into North and South regions. Mean accuracies for each dataset were derived using a 10-fold cross-validation to determine if the accuracies of the fragility models are affected by the compound effect of earthquake and tsunami.

## 9. Conclusions

### 9.1 Main findings and limitations

We presented a first attempt to quantifying structural vulnerability of port industries to tsunami impacts by developing a damage database for port structures and constructing damage fragility functions for various port industries. We were able to collect damage data for more than 5000 port structures and produce damage fragility functions for eight main port industries. Through the interpretations of our damage assessment and statistical analyses of our fragility model, a number of significant findings have emerged from this study:

1. Energy-related and warehousing and distribution industries showed relatively higher resistance to tsunami loads, whereas chemical, cargo handling and construction materials industry appeared to be more vulnerable.

2. Using our proposed damage classification scheme, our fragility models were able to reproduce damage with prediction accuracies of up to 75%, which outperforms models created using aggregated building damage data from MLIT (Leelawat et al., 2014).





3. Pre-tsunami earthquake activities have an influence on port structural damage. It is unavoidable that the compound effects of ground shaking and liquefaction are captured in the damage data, and unaccounted for in the process of developing fragility functions. However, ground shaking appears to influence building damage at lower damage states.

We are also aware of other limitations of this study. One of the limitations which has repeatedly surfaced in our findings is that inundation depth alone is not sufficient to explain the damage observed in port industries. Key threshold depths were difficult to capture for all industries which suggests that by only using inundation depth as a predictor, the fragility models may underestimate the levels of damage sustained by port structures. The models can be further refined by considering other measures of damage such as other tsunami flow characteristics (e.g. velocity, hydrodynamic force), debris impacts or the effects of shielding.

**9.2 Future use of damage database and recommendations**

This study presents an array of potential applications in future port damage studies. First and foremost, a new damage classification scheme was proposed to characterise damage to port structures. This scheme is transferable to other study sites for damage assessment and can be applied to damage assessments through ground survey, photographic interpretation, remote sensing and machine learning techniques. Secondly, we outlined a reproducible method for damage assessment in place of an actual ground survey, especially since this assessment was performed years after the event. The manual assessment allowed us to capture damage details from a side-profile, which otherwise would have been missing from automated techniques such as change detection in remote sensing imagery. In addition, the damage database can also be used in future work to investigate the influence of different parameters such as tsunami flow characteristics, construction characteristics and etcetera on the damage observed. Last but not least, our findings, quantified through the development of fragility functions, can be used to estimate damage to port structures in future tsunami events. They can also be used to motivate improvement in structural designs, tsunami mitigation measures as well as current methods of damage assessment. However, caution must be exercised when applying these models outside of Japan as structural integrity differs from place to place, though we expect that there would be less regional variability for port industries as compared to building codes in houses and commercial buildings.

We invite and provide recommendations for potential users to expand the database and improve the predictive ability of the existing fragility models:

1. Expand the database by collecting damage data from other events and improve the quality of the database by providing more details on the (i) origin of tsunami, (ii) coastal morphological setting, and (iii) method of data collection.

2. Perform tsunami simulation to collect other intensity measures such as velocity and hydrodynamic force.

3. Study the performance of buildings and port infrastructure separately. This would, however, require a larger dataset than presented in this study because fragility models built on smaller sample sizes tend to have greater uncertainty.



**Data availability**

The database provides a comprehensive inventory of port structures and their associated damage in the 2011 Great East Japan
tsunami. The database is available through an unrestricted data repository (DR-NTU) hosted by Nanyang Technological
University (https://doi.org/10.21979/N9/OTZMT1) (Chua et al., 2020). A database guide is provided in the supplementary.

**Author contribution**

CTC designed the study, collected all data and information, performed all statistical analysis and prepared the manuscript.
ADS provided direction for conceptualisation and advice on paper structure. AS provided the original MLIT damage data and
provided guidance on the development of fragility functions. LL and KP provided advice on structural response and tsunami
behaviour. DL provided advice for statistical analysis and development of fragility functions. IC provided advice on building
damage assessment and development of damage database. TC provided advice for statistical analysis and developed code for
bootstrapping techniques. AC assisted in the development of the damage database. SJ and NW provided general direction of
paper. All authors contributed to the scientific discussion of the methods and results, as well as the editing of the manuscript.

**Competing interests**

The authors declare no competing interests.

**Acknowledgements**

This research was supported by the Earth Observatory of Singapore via its funding from the National Research Foundation
Singapore and the Singapore Ministry of Education under the Research Centres of Excellence initiative. This work comprises
EOS contribution number 329. The project was funded by SCOR Reinsurance Asia-Pacific. We are grateful for the support
and advice we have received from Paul Nunn (SCOR Global P&C) and Nigel Winspear (formerly SCOR Global P&C). This
work formed part of the PhD study of CTC, who received funding from the Nanyang Research Scholarship. This study was
supported in part by the facilities and staff at the International Research Institute of Disaster Science (IRIDeS, Tohoku
University). Special thanks go to Professor Fumihiko Imamura, the director of the International Research Institute of Disaster
Science, for supporting and hosting the PhD student in IRIDeS. AS and KP were funded and supported by Tokio Marine &
Nichido Fire Insurance Co., Ltd. and Willis Research Network (WRN). We would also like to thank Janneli Lea Soria, Stephen
Chua and Jedrzej Majewski for providing feedback on the organisation of the manuscript.



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
