# Peer review of "Tsunami damage to ports: Cataloguing damage to create fragility functions from the 2011 Tohoku event"

_Natural Hazards and Earth System Sciences, 2020_

## Referee Comment (RC1) · Anonymous Referee #1 · 25 Dec 2020

The work extended the MLIT database to include more than 5000 port structures pertaining to eight industries. The tsunami damage fragility functions for structures in each industry were obtained with reasonable accuracy. The manuscript is well written and is a valuable contribution to the field of tsunami damage to port structures. The reviewer has minor comments, which the authors may find helpful.

L167: Although the structures are categorised based on industries, it is relevant to mention the structure/building characteristics (steel, concrete, reinforced, components, heights, shapes, etc) specific to each industry in this study. The reviewer feels that the structure/building type in each industry is not clearly described, especially the structural

components and physical features that help to sustain the tsunami impact. The common structural/physical features of structures in a specific industry should be assessed as in lines 374-387.

L175: Since this study uses the maximum inundation depth as the intensity measure, is there any evidence in the literature showing the link between damage to structures and the maximum inundation depth?

L279: The distribution skewed towards the left or right?

L286: Possible reasons for the outliers?

---

## Referee Comment (RC2) · Patricio A. Catalan (Referee) · 23 Mar 2021

In this work, Chua an collaborators present in a very clear and organized manner, the first assessment and proposal of tsunami fragility curves for port industries, hence following the conceptual approach that has been used for other types of buildings to characterize damage. Based on an extensive and well curated dataset from the impact of the Tohoku tsunami, and well sustained methods, they not only propose the fragility curves but also a measure of their accuracy. Moreover, the resulting data also allows them to explore inferring distinguishing between tsunami effects and prior sources of damage.
I found the work very well written, and it flows in a clear and very well organized manner. In general terms, I think it is on a very close state, suitable for publication and it could be an influential work.

With this in mind, I would like to explore a couple of aspects that appear to be bit loose on the article. The justify my "minor revisions" suggestion:

1. It is not mentioned whether tsunami damage by debris or others has been filtered or identified. This issue is particularly present at port facilities, where ships, containers, trucks and other materials relevant to port industries can impact structures. I have heard of work related that has attempted to quantify its impact (by Kentaro Kumagai, for instance).

The reason I mention this is that it could explain some of the features present in data. For instance, in Fig. 5 it is remarkable that DS4 levels can be attained by very small water depths, whilst no DS3 or DS2 states are present. This defies the "ordered" notion of damage that is the basis for the analysis.

Its clarification can also help in explaining some of the later results. The authors point in the paragraph L397-L409 to two main causes for the results: A design oriented to higher standards, that could help in explaining low damage at larger depths; and/or that there is a decorrelation with inundation depth. This is understood that damage is less dependent on water depth alone (at least it is implied in the text. The closing sentence leaves many doors opened and it is inconclusive), though it can be as well interpreted in the opposite way.

Interestingly, incorporating debris can also help in clarifying some of the other features in the data. For example: the authors analyze the variability or accuracy of their results. Typically, they observe that some industries have less data and that is the reason for the low accuracy (smaller data size, which is mentioned once or twice in the text) . However, in reviewing the data, it looks to me that the industries with less accuracy are characterized by a narrow range of depths (for instance, Warehousing does not

exceed 6.0m) whilst having a somewhat uniform distribution of damages. That is, the frequency count of DS1 to DS4 is relatively uniform. So, more damage can be found for a narrow range of small depths. The authors note this explicitly in Line 466, but they chose to use other explanation routes. To me, this also points to the decoupling of damage and flow depth, and could point to other sources of damage that have not been accounted for.

I would recommend the authors to expand a little bit on this regard, and explore its potential effect on their analysis. I think it is a relevant caveat of these studies, because what we see is the end result, and the chain of events that lead to the damage is often absent. It is quite the leap of faith to assume that this is only due to the tsunami hydrodynamics alone, and even from them, just to the water depth.

I have a couple of other suggestions:

Fig 5: The caption is a bit confusing, because it mentions outliers associated with damage first, but then they are related with water depths. At this point the relation is not established.

Figure 7: Perhaps in addition to the frequency count, use percentages. That would allow to compare more clearly among industries.

Line 411 mentions "mean value" but then L418 refers to the median. I tend to think we are referring to median. Please check for consistency throughout the text.

Fig 8: Perhaps gridlines would help to compare among subplots.

---

## Author Comment (AC1) · 21 Apr 2021

The following are our response to the reviewer's specific comments.

**L167: Although the structures are categorised based on industries, it is relevant to mention the structure/building characteristics (steel, concrete, reinforced, components, heights, shapes, etc) specific to each industry in this study. The reviewer feels that the structure/building type in each industry is not clearly described, especially the structural components and physical features that help to sustain the tsunami impact. The common structural/physical features of structures in a specific industry should be assessed as in lines 374-387.**

Thank you for the highly relevant comment. We recognise and agree with the reviewer that the structural components and physical features determine the vulnerability of the industry to tsunami impacts. We have included in Table 2, some of the common infrastructure that can be found in each of these industries, as well as a brief description of common physical assets and construction types as per Lines 168-175 in the corrected manuscript. We hope our revision is satisfactory for the reviewer.

Corrected manuscript:
[Line 168-175] Buildings in port industries commonly include administrative offices, control and maintenance buildings, warehouses and cold storage. Industrial buildings are typically of steel or concrete construction. On the other hand, the types of port infrastructure are diverse - ranging from small transformers to large loading cranes. Some common infrastructure found in each industry are listed in Table 2, adapted from the descriptions provided by the AIR Construction and Occupancy Class Codes (AIR Worldwide, 2019). Because of their diversity, port infrastructure vary widely in their construction and unlike buildings, it is extremely challenging to classify them according to their construction nature. It is interesting to note, however, that several industrial infrastructure are installed in support structures or housed in buildings. In the petrochemical industry, for example, oil and gas are commonly stored in steel or concrete silos and tanks.

Refer to Table 2 in Line 177 and in the Appendix.

**L175: Since this study uses the maximum inundation depth as the intensity measure, is there any evidence in the literature showing the link between damage to structures and the maximum inundation depth?**

We thank you for the opportunity to consider this question. There has not been a consensus on which parameters or rather tsunami intensity measure (TIM) provides the best explanation for damage. There are a number of papers in literature that have evaluated the relative influence of different parameters on building damage (e.g. Macabuag et al., 2016; Song et al., 2017). We are aware that damage to structures are attributed to a combination of many factors and not just inundation depth alone.

Maximum inundation depths are one of the common measures of tsunami damage in literature (e.g. Leone et al., 2011; Suppasri et al., 2013) as they are more easily estimated from field survey after tsunami events as compared to simulated flow values, as pointed out in Line 175-177 [Lines 183 – 185 in corrected manuscript]. For those reasons, we have therefore chosen to work with maximum inundation depths. In this manuscript, our main intention is to create a damage database with primary data. We welcome future users of the damage database to expand beyond using inundation depth as a measure of damage.

Should the reviewer's main concern be on using maximum values, we acknowledge the possibility of damage occurring before inundation reaches maximum depth. This was also addressed in Suppasri et al. (2019), where they found that the critical value for damage may not be at maximum

inundation depths or velocities. We are currently working on a second paper which follows up on the present work, where we evaluate the use of non-maximum inundation values of depth and velocity to explain the damage observed.

**L279: The distribution skewed towards the left or right?**

Thank you for the question. We have clarified this in the text: distribution skewed towards the right (i.e. with a long right tail and a mean to the right of the mode).

**L286: Possible reasons for the outliers?**

The outliers here refer to the inundation depths. The damage data (and hence inundation depths) were collected across different ports in the Tohoku region and therefore, the most plausible explanation for the outliers is that the areas covered in our dataset did not cover the missing depth range. We have removed the description on outliers in Fig.5 caption to avoid the confusion for readers (also pointed out by Reviewer 2), because it has little relevance to the rest of the manuscript.

Corrected manuscript:
[Line 292] Fig. 5. Histograms of each damage state. Distribution of damage data indicates non-normality and DS 1 accounts for the majority of the dataset.

We hope that our responses clarify your concerns. We thank you for your suggestions and for taking the time to review this manuscript. We are happy to address any other questions that you might have.

**References**

AIR Worldwide: AIR Construction and Occupancy Class Code, https://docs.air-worldwide.com/Validation/5.0/index.htm#Exposure_Data/Industrial_Facility_Occupancies.htm, last access: 05 April 2021, 2019.

Leone, F., Lavigne, F., Paris, R., Denain, J. C. and Vinet, F.: A spatial analysis of the December 26th, 2004 tsunami-induced damages: Lessons learned for a better risk assessment integrating buildings vulnerability, *Applied Geography*, 31, 363-375, https://doi.org/10.1016/j.apgeog.2010.07.009, 2011.

Macabuag, J., Rossetto, T., Ioannou, I., Suppasri, A., Sugawara, D., Adriano, B., Imamura, F., Eames, I. and Koshimura, S.: A proposed methodology for deriving tsunami fragility functions for buildings using optimum intensity measures, *Natural Hazards*, 84, 1257-1285, https://doi.org/10.1007/s11069-016-2485-8, 2016.

Song, J., De Risi, R. and Goda, K.: Influence of flow velocity on tsunami loss estimation, *Geosciences*, 7, 114, https://doi.org/10.3390/geosciences7040114, 2017.

Suppasri, A., Mas, E., Charvet, I., Gunasekera, R., Imai, K., Fukutani, Y., Abe, Y. and Imamura, F.: Building damage characteristics based on surveyed data and fragility curves of the 2011 Great East Japan tsunami, *Natural Hazards*, 66, 319-341, https://doi.org/10.1007/s11069-012-0487-8, 2013.

Suppasri, A., Pakoksung, K., Charvet, I., Chua, C. T., Takahashi, N., Ornthammarath, T., Latcharote, P., Leelawat, N., and Imamura, F.: Load-resistance analysis: an alternative approach to tsunami damage assessment applied to the 2011 Great East Japan tsunami, *Nat. Hazards Earth Syst. Sci.*, 19, 1807–1822, https://doi.org/10.5194/nhess-19-1807-2019, 2019.

**Appendix – Corrected Manuscript (Table)**

**Table 2.** Proposed classification for port activities found in the Tohoku region.

| | Industry type | Description of port activities |
|---|---|---|
| Maritime industries | Cargo handling industry | Cargo handling services such as loading and unloading of ships (stevedoring) as well as the handling of cargo on shore.

**Typical infrastructure:** Loading and gantry cranes, storage yards, storage sheds, tanks, chillers and warehouses (buildings). |
| | Warehousing and distribution | Cold storage, warehousing and logistics support.

**Typical infrastructure:** Storage sheds, tanks and silos. |
| Non-maritime port-related industries | Chemical industry | Bulk chemical production e.g. alkane, propane and fertilisers.

**Typical infrastructure:** Distillation towers, tanks, silos, conveyors, pipes, pumps, compressors, reactors, vessels, wastewater treatment systems, chemical separation columns, substations and open frame structures. |
| | Construction materials industry | Concrete and cement manufacturing. Asphalt and wood processing.

**Typical infrastructure:** Rotary kiln/furnace, coal storage, grinders, mills, pre-heating towers, coolers, tanks, silos, conveyors, sorters and stackers. |
| | Energy-related industry | Coal power generation. Electric power generation and distribution.

**Typical infrastructure:** Mills, power plants, substations, transformers, chimneys, boilers, generators, cooling towers, turbines, condensers, pumps and electricity transmission towers. |
| | Food industry | Seafood processing and food packaging. Feed manufacturing.

**Typical infrastructure:** Ovens, cold storage (buildings), freeze dryers, tanks, mixers, conveyors, boilers and vessels. |
| | Manufacturing industry | Metal and alloy products. Plywood and paper products.

**Typical infrastructure:** Grinders/refiners, chimneys, furnaces, silos, tanks, screens, conveyors, cranes, mills and rollers. |
| | Petrochemical industry | Oil depots, reserves and refineries.

**Typical infrastructure:** Furnaces, distillation towers, crackers, compressors, condensers, vessels, tanks, silos, pipelines, |

---

## Author Comment (AC2) · 21 Apr 2021

Please find our responses and corrections to your suggestions below.

**It is not mentioned whether tsunami damage by debris or others has been filtered or identified. This issue is particularly present at port facilities, where ships, containers, trucks and other materials relevant to port industries can impact structures. I have heard of work related that has attempted to quantify its impact (by Kentaro Kumagai, for instance).**

**The reason I mention this is that it could explain some of the features present in data. For instance, in Fig. 5 it is remarkable that DS4 levels can be attained by very small water depths, whilst no DS3 or DS2 states are present. This defies the "ordered" notion of damage that is the basis for the analysis.**

**Its clarification can also help in explaining some of the later results. The authors point in the paragraph L397-L409 to two main causes for the results: A design oriented to higher standards, that could help in explaining low damage at larger depths; and/or that there is a decorrelation with inundation depth. This is understood that damage is less dependent on water depth alone (at least it is implied in the text. The closing sentence leaves many doors opened and it is inconclusive), though it can be as well interpreted in the opposite way.**

**Interestingly, incorporating debris can also help in clarifying some of the other features in the data. For example: the authors analyze the variability or accuracy of their results. Typically, they observe that some industries have less data and that is the reason for the low accuracy (smaller data size, which is mentioned once or twice in the text). However, in reviewing the data, it looks to me that the industries with less accuracy are characterized by a narrow range of depths (for instance, Warehousing does not exceed 6.0m) whilst having a somewhat uniform distribution of damages. That is, the frequency count of DS1 to DS4 is relatively uniform. So, more damage can be found for a narrow range of small depths. The authors note this explicitly in Line 466, but they chose to use other explanation routes. To me, this also points to the decoupling of damage and flow depth, and could point to other sources of damage that have not been accounted for.**

**I would recommend the authors to expand a little bit on this regard, and explore its potential effect on their analysis. I think it is a relevant caveat of these studies, because what we see is the end result, and the chain of events that lead to the damage is often absent. It is quite the leap of faith to assume that this is only due to the tsunami hydrodynamics alone, and even from them, just to the water depth.**

We thank the reviewer for bringing up an important point and providing insightful comments. Unfortunately, through our visual interpretation of spatial sources (such as Google StreetView), we did not assess and found it challenging to identify debris impact at a building level. However, we do agree with the reviewer and acknowledge that debris impact is a potential (and alternative) cause of some of the damage observed. We have looked into available work on the topic and included some of our interpretation in this regard in our revised manuscript.

Corrected manuscript:
[Lines 394 –408] Other factors such as debris impact and proximity of the structure to the shoreline should not be discounted when considering differences in the response of each industry to tsunami impacts. Tsunami-borne debris can contribute significantly to structural damage. This issue is particularly present in port facilities, where ships, containers, mobile equipment, construction materials such as wood logs and concrete objects can impact on structures. Port structures are typically of more robust construction and therefore, they act as barriers in the path of debris motion for as long as inundation depth is lower than the structure height (Reese et al., 2007; Naito et al., 2014). As a result, they are more likely to be subjected to damage from debris impact (Charvet et

al., 2015). While debris impact is location-specific and does not affect all areas in the same ways, some industries may be more susceptible to debris impact than others. For example, in cargo handling and construction materials industries, where mobile large objects such as containers and wood logs are stored in open yards, there is a higher concentration of potential debris and therefore, a higher debris delivery potential (Naito et al., 2014). Kumagai (2013) surveyed the post-mortem dispersal of containers after the 2011 Tohoku event and found that containers, which were not washed out to sea, were mostly dispersed within the terminals where they were located in. Many of these containers were also found to be concentrated around buildings surrounding the container yards without travelling further inland (Kumagai, 2013; Naito et al., 2014), which suggests that damage sustained to structures within these facilities are more likely a consequence of the combined effect of debris impact and tsunami flow than hydrodynamic force alone.

[Lines 441 – 451] These findings can alternatively be justified by the effects of debris impact. A couple of studies (e.g. Charvet et al., 2015; Macabuag et al., 2015) have found the inclusion/omission of debris impact to have an effect on fragility models. Macabuag et al. (2015) demonstrated that models that include regression parameters considering debris impact have a better fit (statistically more significant) than models that do not. The authors also argued that the omission of debris information will likely introduce systematic bias to the fragility models. In this study, debris impact has not been explicitly considered in the development of fragility models, though it could be a source of uncertainty in our fragility models. Intuitively, structures that were damaged by debris would fall into higher damage states and likely experienced higher tsunami intensity values (i.e. depth and velocity). By neglecting debris impact, it is unsurprising that confidence intervals tend to widen towards higher depth values for DS 3 and DS 4 (Fig. 8). Similarly, by neglecting debris information, fragility functions derived for industries, such as cargo handling and construction materials industries, that are more heavily impacted by the debris-related damage are expected to have greater uncertainties.

[Lines 509 – 510] Second-order factors beyond flow regime such as debris impact and proximity to the shoreline could also have an effect on model accuracies.
* * *
**Fig 5: The caption is a bit confusing, because it mentions outliers associated with damage first, but then they are related with water depths. At this point the relation is not established.**

Thank you for pointing this out. We have removed the description on outliers in Fig.5 caption to avoid the confusion for readers, as it has little to no relevance to the rest of the manuscript.

Corrected manuscript:
[Line 292] Fig. 5. Histograms of each damage state. Distribution of damage data indicates non-normality and DS 1 accounts for the majority of the dataset.
* * *
**Figure 7: Perhaps in addition to the frequency count, use percentages. That would allow to compare more clearly among industries.**

Thank you for the suggestion. The changes have been incorporated in Fig. 7. as shown in the appendix below.
* * *
**Line 411 mentions "mean value" but then L418 refers to the median. I tend to think we are referring to median. Please check for consistency throughout the text.**

Thank you for pointing this out, changes have been made in Line 432 in the corrected manuscript.

Corrected manuscript:
[Line 432] … confidence intervals around the  median of the resulting probabilities.

**Fig 8: Perhaps gridlines would help to compare among subplots.**

Thank you for the suggestion. Gridlines have been incorporated into our subplots in Fig. 8, and accordingly in Fig. 11 (refer to Appendix).

We hope that our responses and corrections satisfy and address your concerns. We thank you again for your time and believe that your suggestions have helped improve the quality of this manuscript. We are happy to address any other questions that you might have.

**References**

Charvet, I., Suppasri, A., Kimura, H., Sugawara, D. and Imamura, F.: A multivariate generalized linear tsunami fragility model for Kesennuma City based on maximum flow depths, velocities and debris impact, with evaluation of predictive accuracy, Natural Hazards, 79, 2073-2099, https://doi.org/10.1007/s11069-015-1947-8, 2015.

Kumagai, K.: Tsunami-induced Debris of Freight Containers due to the 2011 off the Pacific Coast of Tohoku Earthquake. Journal of Disaster FactSheets, 1-25, 2013.

Macabuag, J., Rossetto, T., Ioannou, I., & Eames, I.: Investigation of the effect of debris-induced damage for constructing tsunami fragility curves for buildings. Geosciences, *8*(4), 117, https://doi.org/10.3390/geosciences8040117, 2018.

Naito, C., Cercone, C., Riggs, H. R., & Cox, D.: Procedure for site assessment of the potential for tsunami debris impact. Journal of Waterway, Port, Coastal, and Ocean Engineering, 140(2), 223-232, 2014.

Reese S., Cousins W.J., Power W.L., Palmer N.G., Tejakusuma I.G., Nugrahadi S.: Tsunami Vulnerability of buildings and people in South Java—field observations after the July 2006 Java tsunami. Natural Hazards and Earth System Sciences, 7, 573–589, https://doi.org/10.5194/nhess-7-573-2007, 2007.

**Appendix – Corrected Manuscript (Figures)**

[Figure]

**Fig. 7.** Data attributes of the port industries affected by the 2011 Great East Japan tsunami.

[Figure]

**Fig. 8.** Fragility curves with 95% confidence bands for port industries identified in this study. Chemical, cargo handling and construction materials industries appear to be more vulnerable to tsunami inundation depths, while petrochemical and warehousing and distribution industries have lower damage probabilities for the same inundation depths. Wider confidence bands imply greater variability in uncertainty and could be results of smaller sample sizes.

[Figure]

**Fig. 11.** Fragility functions developed for manufacturing industry in (a) North Tohoku, (b) South Tohoku as well as food industry in (c) North Tohoku and (d) South Tohoku. To evaluate the effects of preceding earthquake damage on overall damage assessment, datasets for each industry were divided into North and South regions. Mean accuracies for each dataset were derived using a 10-fold cross-validation to determine if the accuracies of the fragility models are affected by the compound effect of earthquake and tsunami.